# Beta-Arrestin 1 Deficiency Enhances Host Anti-Myeloma Immunity Through T Cell Activation and Checkpoint Modulation

**DOI:** 10.3390/ijms262311478

**Published:** 2025-11-27

**Authors:** Jian Wu, Xiaobei Wang, Shaima Jabbar, Niyant Ganesh, Emily Chu, Vivek Thumbigere Math, Lindsay Rein, Yubin Kang

**Affiliations:** 1Division of Hematologic Malignancies and Cellular Therapy, Department of Medicine, School of Medicine, Duke University Medical Center, Durham, NC 27710, USA; jw731@duke.edu (J.W.); shaimajabbar1@gmail.com (S.J.);; 2Division of Periodontology, Department of Advanced Oral Science & Therapeutics, University of Maryland School of Dentistry, Baltimore, MD 21201, USA; nganesh1@umaryland.edu (N.G.);; 3Department of Biomaterials and Regenerative Dental Medicine, University of Maryland School of Dentistry, Baltimore, MD 21201, USA

**Keywords:** beta-arrestin 1, multiple myeloma, host immunity, T cell activation, immune checkpoint, PD-1, tumor microenvironment

## Abstract

*Beta-arrestin 1* (*ARRB1*) is a multifunctional adaptor protein that regulates diverse signaling pathways beyond its canonical role in G-protein-coupled receptor desensitization. While *ARRB1* has been implicated in cancer progression, its role in modulating host immunity against multiple myeloma (MM) remains unexplored. Here, we demonstrate that host *ARRB1* deficiency significantly enhances anti-myeloma immunity and prolongs survival in a syngeneic murine MM model. Using Vk*MYC myeloma cells transplanted into wild-type and *ARRB1* knockout mice, we show that *ARRB1* deficiency in the host microenvironment promotes robust T cell infiltration and activation while reducing immunosuppressive myeloid populations. Notably, *ARRB1* knockout mice exhibited markedly decreased programmed cell death protein-1 (*PD-1*) expression on both T cells and myeloid-derived suppressor cells, indicating reduced immune exhaustion. Furthermore, *ARRB1* deficiency conferred protection against myeloma-induced bone disease, suggesting a dual role in immune regulation and bone homeostasis. These findings establish ARRB1 as a critical negative regulator of host anti-myeloma immunity and identify it as a potential therapeutic target for enhancing immunotherapy efficacy in MM.

## 1. Introduction

Multiple myeloma (MM) remains an incurable plasma cell malignancy characterized by profound immune dysfunction and a highly immunosuppressive tumor microenvironment [1,2]. Despite significant advances in treatment, including immunomodulatory drugs, proteasome inhibitors, monoclonal antibodies, and T cell-targeted immunotherapy, virtually all patients relapse due to drug resistance and immune evasion [3]. Understanding the molecular mechanisms that govern host anti-myeloma immunity is, therefore, critical for developing more effective therapeutic strategies.

The tumor microenvironment plays a pivotal role in MM pathogenesis and progression [4]. MM cells orchestrate a complex network of interactions with immune cells, stromal cells, and the bone marrow niche to create conditions favorable for tumor growth while suppressing anti-tumor immunity [5]. Key features of the MM microenvironment include expansion of immunosuppressive cell populations such as myeloid-derived suppressor cells (MDSCs) and regulatory T cells, upregulation of immune checkpoint molecules like *PD-1/PD-L1*, and production of immunosuppressive cytokines [6,7,8,9,10].

*Beta-arrestin 1* (*ARRB1*) belongs to the arrestin family, which includes two “visual” arrestins (arrestin 1 and 4) and two non-visual forms, *ARRB1* and *ARRB2* [11,12]. Originally identified as a negative regulator of G-protein-coupled receptor (GPCR) signaling, *ARRB1* is now recognized as a multifunctional adaptor protein that modulates numerous cellular processes, including proliferation, migration, and survival [13,14]. *ARRB1* has emerged as a critical regulator of immune responses through its ability to scaffold diverse signaling complexes [15]. In the immune system, *ARRB1* has been shown to regulate T cell receptor (TCR) signaling, cytokine production, and myeloid cell differentiation [16,17]. Studies have demonstrated that ARRB1 can modulate the expression and function of key immunoregulatory molecules through its interactions with transcription factors and signaling kinases [18]. However, its specific role in orchestrating host immunity against MM has not been investigated.

We recently provided evidence that *ARRB2* modulates immune checkpoints in the myeloma tumor microenvironment [19]. Given the critical importance of immune checkpoints in MM progression and the success of checkpoint blockade therapies in other malignancies, understanding how *ARRB1* influences these pathways in the context of MM could reveal new therapeutic opportunities. In this study, we investigated the role of host *ARRB1* in anti-myeloma immunity using a well-established syngeneic murine model. Our findings reveal that *ARRB1* deficiency in the host microenvironment dramatically enhances anti-tumor immunity through multiple mechanisms, including increased T cell infiltration and activation, reduced MDSC accumulation, and decreased expression of the exhaustion marker *PD-1*. These results identify *ARRB1* as a critical negative regulator of host anti-myeloma immunity and suggest that targeting *ARRBs* could enhance the efficacy of immunotherapy in MM.

## 2. Results

### 2.1. Host ARRB1 Deficiency Resists Tumor Development and Confers Survival Advantage in Murine Myeloma

To investigate the role of host *ARRB1* in anti-myeloma immunity, we utilized the well-characterized Vk*MYC syngeneic mouse model [20,21,22]. Following intravenous injection of Vk*MYC myeloma cells, all wild-type (WT) littermate recipient mice developed MM as measured by the presence of M-protein on serum protein electrophoresis (SPEP), confirming successful tumor engraftment (Figure 1A). None of the *ARRB1* knockout (KO) recipient mice had detectable M-protein (Figure 1A), indicating resistance to myeloma development in *ARRB1*-deficient mice.

Kaplan–Meier analysis revealed a striking survival advantage in *ARRB1*-deficient hosts. Wild-type mice exhibited rapid disease progression with a median survival of 35 days, consistent with the aggressive nature of this model. In contrast, *ARRB1* knockout mice showed significantly extended survival, with the majority surviving beyond 60 days post-injection (*p* < 0.0001, log-rank test) (Figure 1B). This dramatic improvement in survival suggests that host *ARRB1* plays a critical role in conferring anti-myeloma immunity.

### 2.2. ARRB1 Deficiency Prevents Splenic Myeloma Tumor Infiltration

The spleen serves as a major secondary lymphoid organ and site of extramedullary tumor involvement in MM. We observed significant differences in splenic morphology between wild-type littermates and *ARRB1* knockout tumor-bearing mice. At the endpoint, spleens from wild-type littermate mice were markedly enlarged and irregular, consistent with extensive tumor infiltration and disrupted architecture (Figure 2A). In contrast, spleens from *ARRB1* knockout mice remained significantly smaller and maintained a more normal morphology.

Quantitative analysis confirmed a >6-fold reduction in splenic surface area in *ARRB1* knockout mice compared to wild-type littermate controls (0.23 ± 0.05 cm^2^ vs. 1.43 ± 0.29 cm^2^, *p* < 0.001) (Figure 2B). Additionally, CD138+ myeloma cells in spleen and bone marrow were reduced by 17-fold and 3-fold, respectively, in *ARRB1* knockout mice (Figure 2C,D). This preservation of splenic architecture and significantly lower CD138+ myeloma cells in ARRB1-deficient hosts suggests reduced tumor burden and/or altered immune cell trafficking, consistent with enhanced anti-tumor immunity.

### 2.3. ARRB1 Deficiency Protects Against Myeloma-Induced Bone Disease

MM is characterized by extensive osteolytic bone disease resulting from enhanced osteoclast activity and suppressed osteoblast function [23]. To determine whether host *ARRB1* deficiency alters myeloma-induced bone destruction, we performed microCT analysis of distal femurs from wild-type (WT) and *ARRB1* knockout (KO) mice. Remarkably, *ARRB1* knockout mice showed significant protection against bone loss compared to wild-type littermate controls, as evidenced by representative 3D reconstructions showing increased cortical porosity (highlighted in green) and pronounced cortical thinning in WT femurs, features largely absent in KO mice (Figure 3A).

Quantitative analysis revealed that compared to wild-type, *ARRB1*-deficient hosts maintained higher trabecular bone volume fraction (BV/TV) (Figure 3B) despite tumor challenge. Furthermore, *ARRB1* KO mice displayed increased cortical thickness (Ct.Th; mm) (Figure 3C), along with significantly lower cortical porosity (Ct.Po; %) (Figure 3D), pore number (Po.N; n) (Figure 3E), and pore volume (Po.V; mm^3^) (Figure 3F) compared to WT controls. These findings suggest that *ARRB1* not only regulates immune responses but also modulates the bone microenvironment, potentially through effects on osteoclast differentiation or activity.

### 2.4. Host ARRB1 Deficiency Promotes Anti-Tumor T Cell Responses

To determine the immunological basis for tumor resistance and improved survival in *ARRB1* knockout mice, we analyzed immune cell populations at 28 days post-tumor injection (Appendix A). Flow cytometric analysis revealed significant changes in the immune landscape of *ARRB1*-deficient hosts. At baseline, *ARRB1*-deficient mice showed higher percentages of T cells and CD4+ cells, while the populations of B cells and MDSCs were similar between *ARRB-1* knockout mice and wild-type littermate controls (Figure 4A–D). Most notably, after Vk*myc tumor implantation, we observed substantial increases in both CD4+ and CD8+ T cell populations in the spleens of *ARRB1* knockout mice compared to wild-type littermate controls (Figure 4E).

The expansion of T cell populations was accompanied by a marked reduction in immunosuppressive myeloid cells. Total MDSCs and monocytic MDSCs (M-MDSCs), identified as CD11b^+^Ly6C^hi^ cells, were significantly decreased in the peripheral blood and the spleen of *ARRB1* knockout mice (Figure 4B,F). This reciprocal relationship between effector T cells and suppressive myeloid populations suggests that *ARRB1* deficiency shifts the immune balance toward anti-tumor immunity.

### 2.5. ARRB1 Regulates PD-1 Expression on Immune Cells

Given the importance of immune checkpoints in MM progression, we investigated whether *ARRB1* influenced the expression of *PD-1*, a key marker of T cell exhaustion. Flow cytometric analysis at day 28 revealed striking differences in *PD-1* expression between wild-type littermates and *ARRB1* knockout mice. In the spleen, both CD3+ T cells and CD11b+Gr1+ MDSCs from *ARRB1*-deficient hosts showed significantly reduced PD-1 expression compared to wild-type littermate controls (Figure 5A).

Similar patterns were observed in the peripheral blood, where *PD-1* expression was markedly decreased in T cells and MDSCs in *ARRB1* knockout mice (Figure 5B) after tumor implantation. The reduction in *PD-1* expression was consistent across CD4+ and CD8+ T cell subsets, suggesting a global effect of *ARRB1* on immune checkpoint regulation. These findings indicate that host *ARRB1* promotes immune exhaustion in the context of MM, and its deletion maintains T cells in a more activated, functional state.

## 3. Discussion

Our study reveals a previously unrecognized role for host *ARRB1* as a critical negative regulator of anti-myeloma immunity. Using a syngeneic murine model, we demonstrate that *ARRB1* deficiency in the host microenvironment dramatically improves survival through coordinated effects on immune cell composition, activation status, and checkpoint expression. These findings have important implications for understanding MM immune evasion and developing novel therapeutic strategies.

The profound survival advantage observed in *ARRB1*-deficient hosts highlights the importance of host factors in determining MM outcomes. While previous studies have focused on tumor-intrinsic mechanisms of immune evasion, our results demonstrate that modulating host immune regulatory pathways can significantly impact disease progression. The fact that initial tumor engraftment was similar between wild-type and knockout mice suggests that *ARRB1* primarily affects the adaptive immune response rather than initial tumor seeding.

Our observation that *ARRB1* deficiency promotes T cell expansion while reducing MDSC accumulation reveals a dual mechanism for enhanced anti-tumor immunity. MDSCs are major contributors to immune suppression in MM, producing immunosuppressive factors and directly inhibiting T cell function [24]. The reciprocal changes in T cells and MDSCs suggest that *ARRB1* may coordinate the differentiation or trafficking of these populations. Previous studies have shown that *ARRB1* can modulate myeloid cell development through the regulation of *STAT3* and other transcription factors [25], providing a potential mechanistic link.

The striking reduction in *PD-1* expression on immune cells from *ARRB1* knockout mice represents a novel mechanism for *ARRB1*-mediated immune regulation. *PD-1* upregulation is a hallmark of T cell exhaustion in MM and other cancers, limiting the efficacy of anti-tumor immunity [26]. Our findings suggest that *ARRB1* is required for maintaining high *PD-1* expression, potentially through its scaffolding functions in T cell receptor (TCR) signaling pathways. Previous reports have shown that *ARRB1* can act as a scaffold protein facilitating the activation of key signaling pathways, including *NF-κB*, *MAPK*, and *PI3K-AKT*, which are known to influence *PD-1* expression and T cell exhaustion [27]. This discovery is particularly relevant given the clinical success of *PD-1* blockade in other malignancies and ongoing efforts to enhance checkpoint inhibitor efficacy in MM.

Mechanistically, *ARRB1* serves as an essential molecular scaffold that orchestrates the assembly of complex multi-protein interactions within the cell [28]. These complexes are formed through specific protein–protein interactions, with *ARRB1* playing a central role in uniting various components such as kinases, phosphatases, and adapter proteins [29]. This assembly facilitates efficient signaling propagation within intracellular pathways, critical for sustaining proper cellular functions and enabling cells to respond effectively to external stimuli in a coordinated manner.

Within the immune system, *ARRB1* significantly influences TCR signaling. When the TCR on T cells binds to its specific antigen presented by antigen-presenting cells, it triggers a sequence of signaling events. *ARRB1* modulates these events at several levels, interacting with vital signaling molecules in the TCR pathway to either enhance or inhibit their activities [16]. This modulation directly affects the downstream transcriptional programs that determine T cell fate, influencing the activation of transcription factors responsible for promoting T cell proliferation, differentiation into various subsets (including helper T cells, cytotoxic T cells, or regulatory T cells), and survival.

*ARRB1*’s role extends to myeloid cells, potentially modulating *STAT3* and *IRF8*, two key transcription factors. *STAT3* is involved in numerous cellular processes such as growth, survival, and differentiation, with its activation being critical in myeloid-derived suppressor cells (MDSCs) for their function and development. *ARRB1* may influence pathways that activate *STAT3*, thereby impacting MDSC generation and activity [30,31]. Conversely, *IRF8* is vital for normal myeloid cell development, contributing to lineage identity and the regulation of immune function. *ARRB1* might interact with the signaling cascades that govern IRF8 expression or activity, which can affect the maturation, function, and immune-suppressive capacity of MDSCs, ultimately shaping the overall immune response of the body [32].

The protection against myeloma-induced bone disease in *ARRB1* knockout mice reveals an unexpected connection between immune regulation and bone homeostasis. MM-induced osteolysis results from complex interactions between tumor cells, immune cells, and bone cells [33]. Our results suggest that enhanced anti-tumor immunity in *ARRB1*-deficient hosts may indirectly protect against bone destruction by reducing tumor burden. Alternatively, *ARRB1* may directly regulate osteoclast differentiation or function, as suggested by studies showing *ARRB1* involvement in *RANKL* signaling [34].

From a therapeutic perspective, our findings identify *ARRB1* as an attractive target for enhancing anti-myeloma immunity. Unlike traditional immunotherapies that target single pathways, *ARRB1* inhibition could simultaneously enhance T cell function, reduce immunosuppressive myeloid cells, and decrease checkpoint expression. Our findings suggest several important translational opportunities: First, *ARRB1* expression levels in patient bone marrow biopsies could serve as a predictive biomarker to identify patients most likely to benefit from immunotherapy approaches. Second, the dramatic reduction in *PD-1* expression we observed suggests that *ARRB1* inhibition could enhance the efficacy of *PD-1/PD-L1* checkpoint inhibitors, which have shown limited success in multiple myeloma as monotherapy. Third, given that *ARRB1* deficiency simultaneously enhances T cell function and reduces MDSC populations, *ARRB1* targeting could improve CAR-T cell therapy outcomes by creating a more favorable tumor microenvironment. Currently, there is active development of small molecule inhibitors or degraders targeting *ARRB1*, or strategies to disrupt its protein–protein interactions [35,36]. Cmpd-5 (Compound-5) is a novel allosteric inhibitor that binds to a cryptic pocket on *ARRB1* and *RKN5755* was developed as a selective inhibitor for *ARRB1* [36]. A significant challenge in drug development for *ARRB1* is its structural complexity and diverse roles. This functional diversity makes it difficult to design highly specific inhibitors or degraders without causing off-target effects.

Several limitations of our study warrant discussion. First, while we demonstrate clear effects of host *ARRB1* deficiency, the specific cell types responsible for these effects remain to be determined. Conditional knockout studies targeting *ARRB1* in specific immune populations would help dissect these mechanisms. Second, our model uses a single MM cell line; validation in additional models and human samples would strengthen the translational relevance. Finally, the molecular pathways downstream of *ARRB1* that mediate these effects require further investigation.

Future studies should explore several important questions raised by our findings. How does *ARRB1* regulate *PD-1* expression at the molecular level? What are the key *ARRB1* binding partners that mediate its immunosuppressive effects? Can pharmacological *ARRB1* inhibition recapitulate the phenotype of genetic deletion? Addressing these questions will be critical for translating our findings into therapeutic strategies.

## 4. Materials and Methods

### 4.1. Mice and Animal Care

All animal experiments were conducted in accordance with protocols approved by the Duke University Institutional Animal Care and Use Committee (Protocol A074-23-03, approved 4 July 2023). ARRB1 knockout mice on a C57BL/6 background were generously provided by Dr. Robert Lefkowitz (Duke University) and have been previously characterized [37]. Age- and sex-matched wild-type C57BL/6 littermates were used as controls. Mice were housed under specific pathogen-free conditions with standard light/dark cycles and provided food and water ad libitum.

### 4.2. Vk*MYC Myeloma Cells

Vk*MYC myeloma cells were kindly provided by Dr. Leif Bergsagel (Mayo Clinic in Arizona, Phoenix, AZ, USA). These cells were expanded and propagated in vivo in C57BL/6 mice. The cells were harvested from the spleen and stored in liquid nitrogen. For tumor implantation, the Vk*MYC myeloma cells were thawed, washed and injected into mice within 3 h of thawing [20].

### 4.3. Myeloma Model and Survival Analysis

To establish the syngeneic MM model, 1 × 10^6^ Vk*MYC cells were injected intravenously via the tail vein into 8–10-week-old wild-type or *ARRB1* knockout mice [20]. Tumor burden was monitored weekly by serum protein electrophoresis (SPEP) to detect M-protein. Mice were monitored daily for signs of morbidity and euthanized when moribund according to Institutional Animal Care and Use Committee (IACUC) guidelines. Most of the mice had natural death from disease rather than being euthanized for humane endpoints. Survival curves were generated using the Kaplan–Meier method and compared using the log-rank test.

### 4.4. Flow Cytometry Analysis

Single-cell suspensions were prepared from spleens and bone marrow at the indicated timepoints. Red blood cells were lysed using ACK buffer. Cells were pre-incubated with anti-mouse CD16/32 (Fc block) before staining with fluorochrome-conjugated antibodies for 30 min at 4 °C. The following antibodies were used: anti-CD3-PE (clone 17A2), anti-CD4-APC-Cy7 (clone GK1.5), anti-CD8-PE-Cy7 (clone 53–6.7), anti-CD11b-PE (clone M1/70), anti-Ly6C-PE-Cy7 (clone HK1.4), anti-Ly6G-FITC (clone 1A8), and anti-PD-1-FITC (clone 29F.1A12). All antibodies were from BioLegend (San Diego, CA, USA) or BD Biosciences (San Jose, CA, USA). Data were acquired on a BD FACSCanto II and analyzed using FlowJo v10.8 software.

Timing of immune analysis was determined based on preliminary kinetic studies of the Vk*MYC model. Day 28 post-injection was selected as the optimal timepoint for immune profiling because it represents the period when wild-type mice have established detectable tumor burden (M-protein positivity) but have not yet reached humane endpoints, allowing for meaningful comparison of immune cell populations between groups. This timepoint also corresponds to maximal immune cell infiltration based on our pilot studies, occurring before the rapid disease progression phase that typically begins around days 32–35 in this aggressive myeloma model.

### 4.5. Micro-Computed Tomography (Micro CT) Analysis

Bone architecture was assessed by micro-computed tomography (μCT) using a Scanco μCT45 system (Scanco Medical AG, Wangen-Brüttisellen, Switzerland) as previously described [38]. Briefly, femurs were scanned at 70 kVp, 76 μA with 6 μm voxel resolution. Digital Imaging and Communications in Medicine (DICOM) files were created from raw data, exported, and calibrated to five known densities of hydroxyapatite (mg HA/cm^3^). Samples were reconstructed three-dimensionally in AnalyzePro 1.0 software (AnalyzeDirect, Overland Park, KS, USA). Samples were oriented and cropped to identify regions of interest (ROIs). Trabecular bone volume fraction (BV/TV), cortical thickness (Ct. Th; mm), cortical porosity (Ct.Po; %), pore number (Po.N; n), and pore volume (Po.V; mm^3^) were calculated from defined ROIs.

### 4.6. Serum Protein Electrophoresis

Serum samples were collected via mandibular vein bleeding and analyzed by agarose gel electrophoresis using the QuickGel™ Chamber (Helena Laboratories, Beaumont, TX, USA). M-protein levels were quantified by densitometry of the monoclonal spike.

### 4.7. Statistical Analysis

Data are presented as mean ± SEM. Statistical comparisons between two groups were performed using an unpaired Student’s *t*-test with Welch’s correction for unequal variances where appropriate. Survival curves were compared using the log-rank (Mantel–Cox) test. *p*-values < 0.05 were considered statistically significant. All analyses were performed using GraphPad Prism v9.0.

## 5. Conclusions

In summary, we identify host ARRB1 as a critical negative regulator of anti-myeloma immunity that promotes immune exhaustion and limits survival. ARRB1 deficiency enhances T cell responses, reduces immunosuppressive myeloid populations, and decreases PD-1 expression, creating an immune microenvironment that is hostile to tumor growth. These findings establish ARRB1 as a novel therapeutic target for enhancing immunotherapy efficacy in MM and potentially other malignancies characterized by immune dysfunction. Our results underscore the importance of host immune regulatory pathways in cancer outcomes and suggest that targeting these pathways could unlock the full potential of cancer immunotherapy.

## Figures and Tables

**Figure 1 ijms-26-11478-f001:**
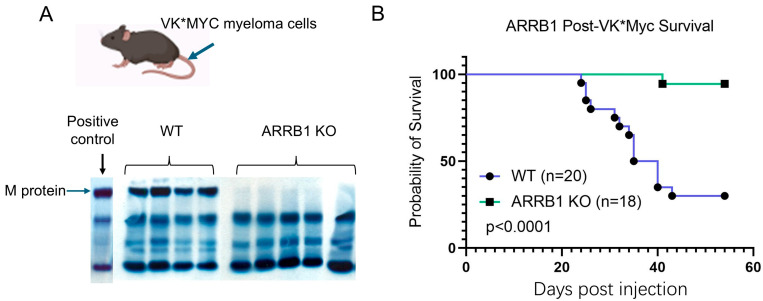
Host ARRB1 deficiency confers survival advantage in syngeneic multiple myeloma mouse model. (**A**) Representative serum protein electrophoresis (SPEP) profiles from wild-type (WT) littermates and *ARRB1* knockout (KO) mice at day 21 following intravenous injection of 1 × 10^6^ Vk*MYC myeloma cells. Arrows indicate M-protein bands. All WT mice (*n* = 18) developed detectable M-protein, confirming successful tumor engraftment. In contrast, no *ARRB1* KO mice (*n* = 18) showed detectable M-protein at this timepoint. (**B**) Kaplan–Meier survival curves comparing WT littermates (blue line) and *ARRB1* KO (green line) mice following Vk*MYC myeloma cell injection. WT littermate mice exhibited rapid disease progression with median survival of 35 days, while *ARRB1* KO mice showed significantly prolonged survival with >80% surviving beyond 60 days (*n* = 18 per group; log-rank test).

**Figure 2 ijms-26-11478-f002:**
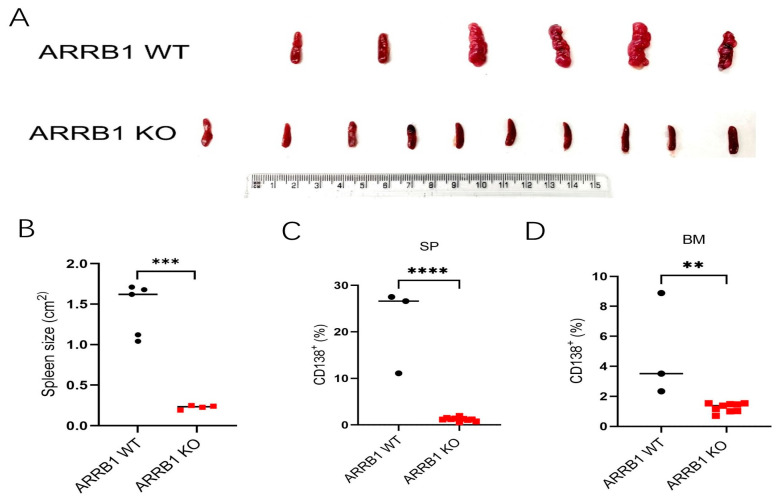
ARRB1 deficiency preserves splenic architecture and reduces tumor burden. (**A**) Representative photographs of spleens harvested at experimental endpoint from WT littermates and ARRB1 KO mice. WT spleens display marked splenomegaly with irregular morphology and visible tumor nodules (**top panel**), while ARRB1 KO spleens maintain smaller size and more normal architecture (**bottom panel**). Scale bar = 1 cm. (**B**) Quantitative analysis of splenic surface area measured using ImageJ 1.45 software. *ARRB1* KO mice showed >6-fold reduction in spleen size compared to WT controls (WT: 1.43 ± 0.29 cm^2^; KO: 0.23 ± 0.05 cm^2^; *n* = 8–10 per group; *** *p* < 0.001, unpaired *t*-test). (**C**,**D**) CD138+ myeloma cells in spleen (**C**) and in bone marrow (**D**) were measured by flow cytometry (** *p* < 0.01, **** *p* < 0.0001, unpaired *t*-test). Data presented as mean ± SEM with individual data points.

**Figure 3 ijms-26-11478-f003:**
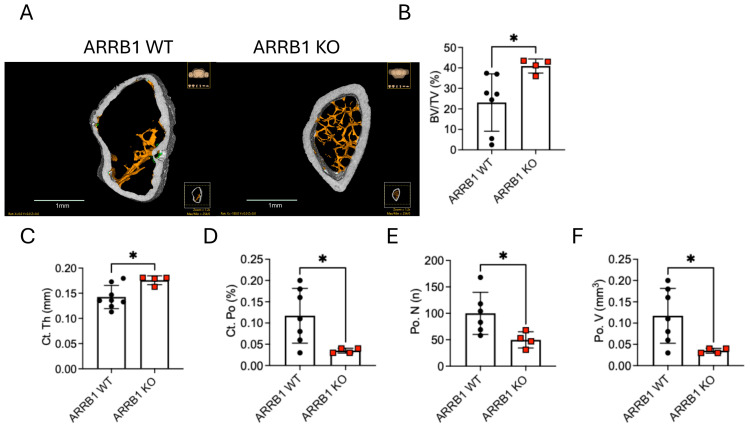
Host *ARRB1* deficiency protects against myeloma-induced bone disease. (**A**) Representative micro-computed tomography (microCT) reconstructions of distal femurs from wild-type (WT) and ARRB1 knockout (KO) mice at the experimental endpoint. WT mice exhibit pronounced cortical thinning and increased porosity (highlighted in green), along with substantial loss of trabecular bone. In contrast, ARRB1 KO mice retain both cortical and trabecular bone integrity. Yellow markings indicate regions of cortical porosity and bone less in WT femurs, demonstrating extensive myeloma-induced osteolytic damage. (**B**) Quantification of trabecular bone volume fraction (BV/TV) reveals significantly preserved bone architecture in ARRB1 KO mice compared to WT controls. (**C**–**F**) Cortical bone analysis shows that ARRB1 KO mice exhibit increased cortical thickness (Ct.Th; mm) (**C**) and significantly reduced cortical porosity (Ct.Po; %) (**D**), pore number (Po.N; n) (**E**), and pore volume (Po.V; mm^3^) (**F**) relative to WT mice. Data are presented as mean ± SEM (*n* = 4–8 per group). * *p* < 0.05 by unpaired two-tailed *t*-test.

**Figure 4 ijms-26-11478-f004:**
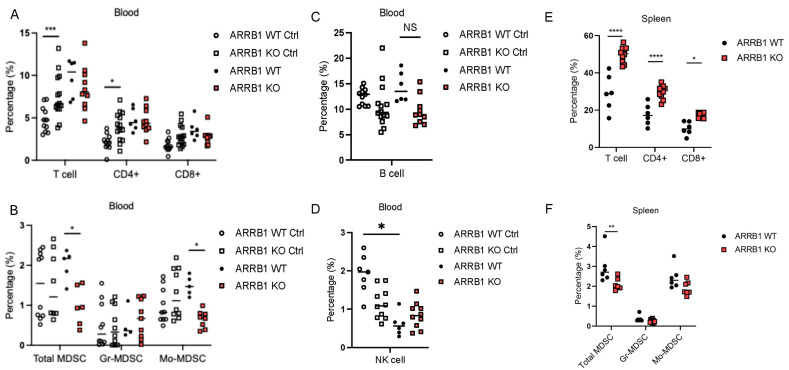
*ARRB1* deficiency enhances T cell responses and reduces immunosuppressive myeloid populations. (**A**–**D**) Flow cytometry quantification of immune cells in peripheral blood before and 28 days post Vk*myc myeloma cell injection. Open circle: WT controls before tumor injection. Open square: *ARRB1* KO mice before tumor injection. Solid circle: WT controls at day 28 post-tumor injection. Solid square: *ARRB1* KO mice at day 28 post-tumor injection (*n* = 8–10 per group; NS: not statistically significant, * *p* < 0.05, *** *p* < 0.001, unpaired *t*-test). (**E**) Flow cytometric quantification of immune cell populations in spleens at day 28 post-injection. *ARRB1* KO mice showed significant expansion of all T cell subsets compared to WT controls (*n* = 8–10 per group; * *p* < 0.05, **** *p* < 0.0001, unpaired *t*-test). (**F**) Representative quantification of monocytic myeloid-derived suppressor cells (M-MDSCs) identified as CD11b^+^Ly6C^hi^ cells. *ARRB1* KO mice demonstrated marked reduction in splenic M-MDSC frequency compared to WT mice (*n* = 8–10 per group; ** *p* < 0.01, unpaired *t*-test). Gates were set based on fluorescence-minus-one controls. Data presented as mean ± SEM with individual data points.

**Figure 5 ijms-26-11478-f005:**
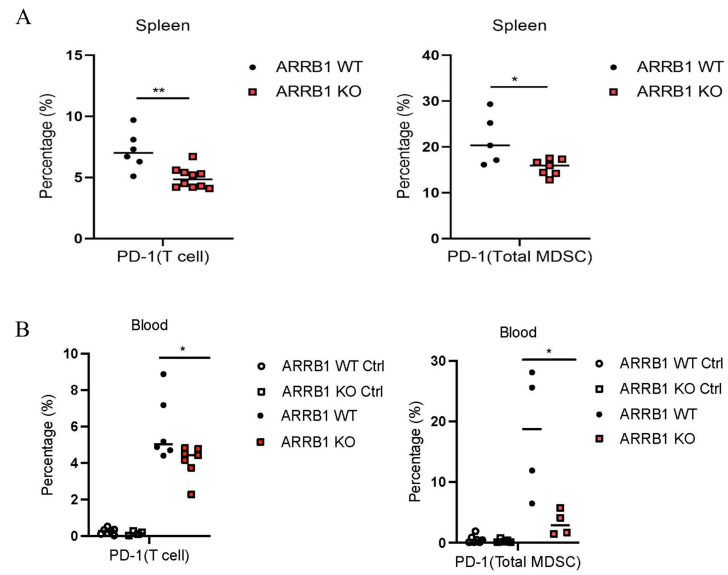
*ARRB1* deletion reduces *PD-1* expression on immune cells in multiple myeloma microenvironment. (**A**) Representative flow cytometry histograms showing *PD-1* expression on splenic immune cells at day 28 post-injection. Left panels: *PD-1* expression on CD3^+^ T cells from WT and *ARRB1* KO mice with isotype control. Right panels: *PD-1* expression on CD11b^+^Gr1^+^ myeloid cells. Numbers indicate percentage of PD-1^+^ cells. Quantification graphs show significant reduction in PD-1^+^ cells in both populations from *ARRB1* KO mice (*n* = 6–8 per group; * *p* < 0.05, ** *p* < 0.01, unpaired *t*-test). (**B**) Similar analysis of peripheral blood cells showing consistent pattern of reduced *PD-1* expression on both CD3^+^ T cells and CD11b^+^Gr1^+^ myeloid cells in *ARRB1* KO mice compared to WT controls. Open circle: WT controls before tumor injection. Open square: *ARRB1* KO mice before tumor injection. Solid circle: WT controls at day 28 post-tumor injection. Solid square: *ARRB1* KO mice at day 28 post-tumor injection. (*n* = 6–8 per group; * *p* < 0.05, unpaired *t*-test). Data presented as mean ± SEM.

## Data Availability

The original contributions presented in this study are included in the article/Appendix A. Further inquiries can be directed to the corresponding author.

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
