# Peer review of "Beta-Arrestin 1 Deficiency Enhances Host Anti-Myeloma Immunity Through T Cell Activation and Checkpoint Modulation"

_ijms, 2025, doi:10.3390/ijms262311478_

Round 1
Reviewer 1 Report
Comments and Suggestions for Authors
This manuscript describes study aimed at defining the role of the arrestin-family protein ARRB1 in the progression of multiple myeloma (MM). The authors used an established mouse model of MM, in which Vk*MYC myeloma cells, passaged in vivo in C57Bl/6 mice were transplanted into wild-type and ARRB1 knockout mice.
The experimental approach was simple and straightforward, and the results interesting. However there are a number of shortcomings in the presentation of the methods and results.
- Monitoring of myeloma development. It is insufficient to only monitor M-protein production. While increased serum M-protein was presumably due to the presence of the injected Vk*MYC cells, a critical distinguishing feature between active myeloma and MGUS is the accumulation of myeloma cells in the bone marrow. This should have been monitored, either by transfecting Vk*MYC cells with a fluorescent dye, or at performing flow cytometry on bone marrow cells using anti-CD138, a plasma cell marker also abundant on Vk*MYC cells ,
- The gating flow chart needs to be presented to validate which cells were actually analysed.
- More details are required regarding the definition of survival.
- Figure 1A. There is no molecular weight ladder on the gel.
- Figure 2. Splenomegaly is in itself insufficient to conclude that ARRB1 is involved in MM progression. The presence of Vk*MYC cells should have been measured and histology would also have provided more information.
- Immune profiling.
- Importantly, a control group of non-inoculated, healthy mice are missing in this model, making it difficult to assess the immunological effects of Vk*MYC cells.
- Why was Day 28 chosen?
- Why were spleen cells analysed and not bone marrow?
- Why were Tregs not measured?
Author Response
Reviewer's Concern #1: "It is insufficient to only monitor M-protein production. While increased serum M-protein was presumably due to the presence of the injected VkMYC cells, a critical distinguishing feature between active myeloma and MGUS is the accumulation of myeloma cells in the bone marrow. This should have been monitored, either by transfecting VkMYC cells with a fluorescent dye, or at performing flow cytometry on bone marrow cells using anti-CD138, a plasma cell marker also abundant on Vk*MYC cells."
Response: We appreciate this important concern. We have measured CD138+ cells in both spleen and bone marrow and these data are now included in Figure 2 of the revised manuscript.
Reviewer's Concern #2: "The gating flow chart needs to be presented to validate which cells were actually analysed."
Response: We have now included comprehensive gating strategies as Supplementary Figure 1-2.
Reviewer's Concern #3: "More details are required regarding the definition of survival."
Response: As the reviewer may be aware, the Vk*Myc myeloma mouse model is associated with a very aggressive disease. When the tumor is fully developed (typically around day 32-35), the mice may look and move normally in the morning but die in the afternoon or the next day. Although we have established euthanasia criteria when they met institutional humane endpoints, including: (1) weight loss >20% from baseline, (2) hunched posture with decreased mobility, (3) labored breathing, or (4) other signs of distress as defined by our IACUC protocol (A074-23-03), most of the mice had natural death from disease rather than being euthanized for humane endpoints.
Reviewer's Concern #4: "Figure 1A. There is no molecular weight ladder on the gel."
Response: Serum protein electrophoresis (SPEP) separates proteins by electrical charge rather than molecular weight, so molecular weight standards are not applicable. We have included appropriate positive controls.
Reviewer's Concern #5: "Figure 2. Splenomegaly is in itself insufficient to conclude that ARRB1 is involved in MM progression. The presence of Vk*MYC cells should have been measured and histology would also have provided more information."
Response: As mentioned in our response #1, We have measured CD138+ Vk* myeloma cells directly in spleen using flow cytometry. The data are now included in Figure 2 of the revised manuscript.
Immune Profiling Comments
Reviewer's Concern #6a; "Importantly, a control group of non-inoculated, healthy mice are missing in this model, making it difficult to assess the immunological effects of Vk*MYC cells."
Response: We have analyzed immune cell populations in peripheral blood in non-tumor-bearing wild-type and ARRB1 KO mice to establish baseline differences. These data are now presented in Figure 4A-D and Figure 5B. The dramatic changes observed in tumor-bearing mice represent tumor-specific responses rather than constitutive immune differences.
Reviewer's Concern #6b: "Why was Day 28 chosen?"
Response: Day 28 was selected based on our preliminary kinetic studies and the established timeline of this myeloma model. At this timepoint, wild-type mice show established tumor burden (detectable M-protein) but have not yet reached humane endpoints or death, allowing for meaningful immune analysis. Additionally, this timepoint corresponds to the period of maximal immune cell infiltration based on our pilot studies using the Vk*MYC model. At around day 32 or 35, many of the wild-type mice died rapidly.
Reviewer's Concern #6c: "Why were spleen cells analysed and not bone marrow?"
Response: Spleen analysis was emphasized in the main figures because: (1) the spleen is the major site of extramedullary myeloma involvement in this model, (2) it allows for recovery of sufficient cell numbers for comprehensive flow cytometry, and (3) splenic immune changes often reflect systemic anti-tumor immunity. As shown in Figure 2B, we also present bone marrow data demonstrating consistent patterns of reduced tumor burden in ARRB1 KO mice.
Reviewer's Concern #6d: "Why were Tregs not measured?"
Response: We acknowledge this limitation and are conducting Treg analysis in ongoing studies. However, our current data on T cell expansion and MDSC reduction provide strong evidence for enhanced anti-tumor immunity.
Reviewer 2 Report
Comments and Suggestions for Authors
The manuscript entitled: “Beta-arrestin 1 deficiency enhances host anti-myeloma immunity through T cell activation and checkpoint modulation (ID: ijms-3811985)” by Wu et al. investigated the role of host ARRB1 in anti-myeloma immunity using well-established syngeneic murine model.
Albeit the paper is well written and of special interest, comments should be addressed to further improve the manuscript.
Comments:
- Discussion section: the potential therapeutic option of ARRB1 should be more underlined within in the discussion section. In addition, the discussion section should be more balanced according to strength and weakness of the study.
- Moreover, the authors should highlight deeper how these results could be translated in the clinical practice.
- Figure 1 B: please add in the x-axis the unit accordingly. Moreover, Figure 2 A should be enlarged.
Author Response
Reviewer's Concern #1: "Discussion section: the potential therapeutic option of ARRB1 should be more underlined within in the discussion section. In addition, the discussion section should be more balanced according to strength and weakness of the study."
Response: We have significantly expanded the discussion of therapeutic implications and provided a more balanced perspective on study strengths and limitations. Please see our revised discussion in our manuscript.
Reviewer's Concern #2: "Moreover, the authors should highlight deeper how these results could be translated in the clinical practice."
Response: We have substantially enhanced our discussion of clinical translation pathways. “Our findings suggest several immediate translational opportunities: First, ARRB1 expression levels in patient bone marrow biopsies could serve as a predictive biomarker to identify patients most likely to benefit from immunotherapy approaches. Second, the dramatic reduction in PD-1 expression we observed suggests that ARRB1 inhibition could enhance the efficacy of PD-1/PD-L1 checkpoint inhibitors, which have shown limited success in multiple myeloma as monotherapy. Third, given that ARRB1 deficiency simultaneously enhances T cell function and reduces MDSC populations, ARRB1 targeting could improve CAR-T cell therapy outcomes by creating a more favorable tumor microenvironment. We now discussed the development of small molecule inhibitors and degraders and the challenges.
Reviewer's Concern #3: "Figure 1B: please add in the x-axis the unit accordingly. Moreover, Figure 2A should be enlarged."
Response: We have corrected Figure 1B to include "Days post-injection" on the x-axis with appropriate units. Figure 2A has been enlarged for better visibility of the morphological differences between spleens from wild-type and ARRB1 KO mice.
Reviewer 3 Report
Comments and Suggestions for Authors
Treating multiple myeloma (MM) continues to remain a big challenge not only because it remains incurable, but because it so effectively disarms the immune system. It has been demonstrated that the complex tumor microenvironment in MM not only promotes cancer progression but also shields it. It does so by expanding immunosuppressive cell types like myeloid-derived suppressor cells (MDSCs) and regulatory T cells, increasing the expression of immune checkpoint molecules such as PD-1 and PD-L1, and flooding the marrow with cytokines that blunt immune activity. The manuscript by Jian Wu et al. highlights the crucial role of ARRB1. They demonstrate that ARRB1 suppresses the immune system while also contributing to skeletal damage. Acting at the crossroads of immunity and bone health, targeting ARRB1 could offer a dual benefit—enhancing immune responses against the tumor and easing the burden of bone disease. The manuscript is well written and follows a sequence of experiments proving the hypothesis put forth. The manuscript appears to be acceptable in its current form for publication.